# Effects of a mindfulness-based intervention and a health self-management programme on psychological well-being in older adults with subjective cognitive decline: Secondary analyses from the SCD-Well randomised clinical trial

Marco Schlosser[1,2], Harriet Demnitz-King[2], Thorsten Barnhofer[3], Fabienne Collette[4,5], Julie Gonneaud[6], Gaël Chételat[6], Frank Jessen[7,8,9], Matthias Kliegel[1], Olga M. Klimecki[10,11], Antoine Lutz[12‡*], Natalie L. Marchant[2‡*], on behalf of the Medit-Ageing Research Group¶

1 Department of Psychology, Faculty of Psychology and Educational Sciences, University of Geneva, Geneva, Switzerland, 2 Division of Psychiatry, Faculty of Brain Sciences, University College London, London, United Kingdom, 3 School of Psychology, University of Surrey, Surrey, United Kingdom, 4 GIGA-CRC In Vivo Imaging, University of Liège, Liège, Belgium, 5 Psychology and Neuroscience of Cognition Research Unit, Faculty of Psychology and Educational Sciences, University of Liège, Liège, Belgium, 6 Normandie Univ, UNICAEN, INSERM, U1237, PhIND "Physiopathology and Imaging of Neurological Disorders", Institut Blood and Brain @ Caen-Normandie, Cyceron, Caen, France, 7 Department of Psychiatry, Medical Faculty, University of Cologne, Cologne, Germany, 8 German Center for Neurodegenerative Diseases (DZNE), Bonn, Germany, 9 Excellence Cluster on Cellular Stress Responses in Aging-Associated Diseases (CECAD), University of Cologne, Cologne, Germany, 10 Swiss Center for Affective Sciences, University of Geneva, Geneva, Switzerland, 11 Clinical Psychology and Behavioral Neuroscience, Faculty of Psychology, Technische Universität Dresden, Dresden, Germany, 12 Eduwell team, Lyon Neuroscience Research Center Inserm U1028, CNRS UMR5292, Lyon 1 University, Lyon, France

‡ AL and NLM share last authorship on this work.
¶ Membership of the Medit-Ageing Research Group is provided in the Acknowledgments.
* n.marchant@ucl.ac.uk (NLM); antoine.lutz@inserm.fr (AL)

## Abstract

### Objectives

Older adults with subjective cognitive decline (SCD) recruited from memory clinics have an increased risk of developing dementia and regularly experience reduced psychological well-being related to memory concerns and fear of dementia. Research on improving well-being in SCD is limited and lacks non-pharmacological approaches. We investigated whether mindfulness-based and health education interventions can enhance well-being in SCD.

### Methods

The SCD-Well trial (ClinicalTrials.gov: NCT03005652) randomised 147 older adults with SCD to an 8-week caring mindfulness-based approach for seniors (CMBAS) or an active comparator (health self-management programme [HSMP]). Well-being was assessed at

**Data Availability Statement:** The data underlying this report are made available on request following approval by the executive committee and a formal data sharing agreement (https://silversantestudy. eu/2020/09/25/data-sharing). The Material can be mobilized, under the conditions and modalities defined in the Medit-Ageing Charter by any research team belonging to an Academic institution, for carrying out a scientific research project relating to the scientific theme of mental health and well-being in older people. The Material may also be mobilized by non-academic third parties, under conditions, in particular financial, which will be established by separate agreement between Inserm and by the said third party. Data sharing policies described in the Medit-Ageing charter are in compliance with our ethics approval and guidelines from our funding body. Data contain potentially identifying or sensitive patient information. To request data, please contact the data access committee via the official project website (https://silversantestudy.eu/2020/09/25/data-sharing).

**Funding:** The SCD-Well Randomised Controlled Trial is part of the Medit-Ageing project funded through the European Union in Horizon 2020 programme related to the call PHC22 "Promoting mental well-being in the ageing population" and under grant agreement No667696. FC was supported by Fonds National de la Recherche Scientifique (FRSFNRS, Belgium). The funders had no role in study design, data collection and analysis, decision to publish, or preparation of the manuscript.

**Competing interests:** GC, FC, OMK, AL, and NLM have received research support from the EU's Horizon 2020 research and innovation programme (grant agreement number 667696). GC has received research support from Inserm, Fondation d'entreprise MMA des Entrepreneurs du Futur, Fondation Alzheimer, Programme Hospitalier de Recherche Clinique, Région Normandie, Association France Alzheimer et maladies apparentées and Fondation Vaincre Alzheimer (all to Inserm), GC and AL have received research support and personal fees from Fondation d'entreprise MMA des Entrepreneurs du Futur. All other authors have declared that no competing interests exist.

baseline, post-intervention, and 6-month post-randomisation using the Psychological Well-being Scale (PWBS), the World Health Organisation's Quality of Life (QoL) Assessment psychological subscale, and composites capturing meditation-based well-being dimensions of awareness, connection, and insight. Mixed effects models were used to assess between- and within-group differences in change.

## Results

CMBAS was superior to HSMP on changes in connection at post-intervention. Within both groups, PWBS total scores, psychological QoL, and composite scores did not increase. Exploratory analyses indicated increases in PWBS autonomy at post-intervention in both groups.

## Conclusion

Two non-pharmacological interventions were associated with only limited effects on psychological well-being in SCD. Longer intervention studies with waitlist/retest control groups are needed to assess if our findings reflect intervention brevity and/or minimal base rate changes in well-being.

## Introduction

Subjective cognitive decline (SCD) describes self-reported worsening of cognitive functioning despite unimpaired performance on objective tests of cognition [1]. Clinical and epidemiological data suggest that older adults with SCD, especially those recruited from memory clinics, are at a higher risk of subsequently developing dementia [2]. The aetiology of SCD is heterogeneous and its phenomenology complex [1]. Although the condition could be an indication of prodromal Alzheimer's disease (AD), which is the most common form of dementia [3], SCD has also been related to other factors (e.g., physical and mental illness, sleep disturbances, personality traits, effects of drugs). Partly due to the heterogeneity of this population and the fact that SCD symptoms frequently remit, there is no consensus on best treatment and management for SCD. Nonetheless, in the absence of effective interventions for curing or treating AD, interest in SCD continues to grow as targeted interventions at this earlier stage could reduce the risk of cognitive decline and progression to AD.

An important aspect of living with SCD is the impact that perceiving increasing cognitive difficulties has on an individual's psychological well-being. The lived subjective experience of individuals with SCD is commonly marked by stress, fear of dementia, anger, and feelings of anxiety and depression [4, 5]. This aspect can be overlooked within research contexts that focus primarily on the maintenance of cognition or the prevention of amyloid deposition. A recent meta-analysis indicated that group psychological interventions moderately increased psychological well-being in SCD (Hedges' g = 0.40; [6]) although none of the included studies, when considered individually, found statistically significant improvements. The authors concluded that existing research on enhancing psychological well-being in SCD is of low quality (e.g., lacking active comparison conditions) and highlighted the striking lack of research on non-pharmacological approaches including lifestyle and mindfulness-based interventions (MBIs).

In line with prior research and theory [7, 8], MBIs have been proposed as a promising strategy for increasing psychological well-being and human flourishing. However, prior to the SCD-Well trial [9], only one study–a small pilot randomised controlled trial (n = 15; [10])– had evaluated the effects of mindfulness training in individuals with SCD. This trial primarily focussed on reaction time and EEG/ERP correlates, change in brain volume, self-reported cognitive complaints, and memory self-efficacy; it did not include measures of psychological well-being or related constructs. Understanding how psychological well-being in SCD, irrespective of its aetiology, could be improved through MBIs remains an important lacuna in this emerging field.

Other promising non-pharmacological interventions for SCD include psychoeducation programmes that provide information on healthy diet, physical exercise, and management of existing health conditions [1]. Strengthening self-efficacy and thus enabling individuals with SCD to live a more active life could be a mechanism by which psychoeducation maintains or improves psychological well-being. A particularly pertinent feature of both MBIs and psychoeducation is their potential to be feasibly implemented in clinical settings. Furthermore, non-pharmacological interventions could empower individuals with SCD to actively learn skills that could enhance their psychological well-being and mental health instead of passively observing how their clinical trajectory unfolds.

Research on the dimensions of psychological well-being has expanded substantially over the past decade, delivering valuable insights into the conditions that predict and contribute to positive functioning (e.g., [7, 11]). To appreciate the conceptual richness of this field and to capture diverse aspects of psychological well-being, we utilised outcome measures derived from three distinct, prominent theoretical models of human flourishing, namely Ryff's theory of well-being [12], the World Health Organisation's conception of psychological quality of life [13], and a recent meditation training-based framework for human flourishing developed by Dahl et al. [7].

Ryff [12] offered the first attempt at providing a unifying theoretical framework for contemporary scientific perspectives on human flourishing. Ryff's influential work was a response to the largely data-driven and atheoretical research on well-being that had hitherto characterised this area. In this model, Ryff aimed to identify the fundamental aspects of positive functioning that could help define what it means to be psychologically well. The Psychological Well-being Scale (PWBS; [14]), which was developed to empirically capture Ryff's proposed dimensions of well-being, is the most cited self-report measure of well-being to date.

The World Health Organisation (WHO) defines quality of life as "individuals' perceptions of their position in life in the context of the culture and value systems in which they live and in relation to their goals, expectations, standards and concerns" [13] and frames quality of life as an aspect of well-being. The WHO Quality of Life (WHOQOL) assessment was developed to capture aspects of quality of life. The introduction of the WHOQOL was a statement of commitment to promoting a genuinely holistic approach to health and health care interventions, echoing the WHO's definition of health as "A state of physical, mental and social well-being, not merely the absence of disease and infirmity" [13].

Dahl et al.'s [7] meditation training-based model of human flourishing integrates insights from neuroscientific and psychological research on well-being with contemplative perspectives. It rests on a skill-based conception of human flourishing, framing dimensions of well-being as trainable capacities. The authors aimed to introduce a set of constructs that could further unify existing theories and interventions in this field while offering a common language to encourage collaboration across related research areas. No self-report measure has yet been developed that was explicitly derived from this model. However, recent research [15] has used

this model to group already published self-report measures into psychometrically sound composites of meditation-based well-being.

The present study aimed to compare the effects of an 8-week MBI adapted for older adults with SCD (caring mindfulness-based approach for seniors; CMBAS) on measures of mental well-being derived from the three approaches described above to a structurally matched health self-management programme (HSMP). We hypothesised that both interventions would improve well-being but that CMBAS would be superior to HSMP, because, based on previous research and theory [7, 16, 17], we predicted embodied meditative practices aimed at deep human flourishing to be a more potent catalyst of well-being than health educational instructions.

## Methods

This study utilised longitudinal data from the SCD-Well randomised controlled trial of the European Union's Horizon 2020-funded Medit-Ageing European project (public name: Silver Santé Study). Detailed information about the recruitment procedures, eligibility criteria, design of the interventions, and assessments can be found in the trial protocol [18].

### Study design

SCD-Well was a multi-center, randomized, controlled, superiority trial with two intervention arms: an 8-week CMBAS and a structurally matched HSMP. Randomisation to one of the two groups was performed at a ratio of 1:1. Participants were assessed at three visits: pre-intervention at baseline (V1), post-intervention (V2), and at follow-up 6 months after randomisation (V3). The primary outcome of the SCD-Well trial was mean change in anxiety symptoms from V1 to V2 [9].

The intervention was delivered at four European sites (Barcelona, Cologne, London, and Lyon). Written informed consent was obtained from all participants after the procedures had been explained to them and prior to participation. The multi-centre SCD-Well trial received ethics approval from the committees and regulatory agencies of all centres: London, UK (Queen Square Research Ethics Committee: n° 17/LO/0056 and Health Research Authority National Health Service, IRAS project ID: 213008); Lyon, France (Comité de Protection des Personnes Sud-Est II Groupement Hospitalier Est: n° 2016-30-1 and Agence Nationale de Sécurité du Médicament et des Produits de Santé: IDRCB 2016-A01298-43); Cologne, Germany (Ethikkommission der Medizinischen Fakultät der Universität zu Köln: n° 17–059); and Barcelona, Spain (Comité Etico de Investigacion Clinica del Hospital Clinic de Barcelona: n° HCB/2017/0062). The SCD-Well trial was performed in accordance with the ethical standards laid down in the 1964 Declaration of Helsinki and its later amendments.

### Participants

A total of 147 older adults (age range: 60 to 91 years in CMBAS; 60 to 87 years in HSMP) were randomised. Participants had no major neurological or psychiatric disorders, and no present or past regular or intensive practice of meditation, were recruited from memory clinics at four European sites, and met the research criteria for SCD proposed by the SCD-I working group [19].

### Interventions

**Caring mindfulness-based approach for seniors (CMBAS).** CMBAS followed the structure of a mindfulness-based stress reduction (MBSR) programme and was tailored to the

needs of older adults with a focus on compassion and loving-kindness meditation. CMBAS also included psychoeducational components that offered participants approaches to deal with cognitive concerns and tendencies to worry in skilful ways. The intervention consisted of eight weekly group sessions of approximately 2 hours, home practice (e.g., guided meditations, informal practices) for 1 hour per day on six days per week, and one retreat day during the sixth week of the intervention that involved 5 hours of practice. CMBAS was delivered to groups of 7 to 12 participants by clinically trained facilitators who had completed training that aligned with the good practice guidelines for mindfulness teachers developed by The Mindfulness Network UK.

**Health self-management programme (HSMP).** HSMP followed the same format and structure as CMBAS, and was matched in administration, duration, and dosage of group meetings including a retreat day with a healthy lunch and topical discussions. HSMP was based on a published manual that included guidance on exercise, stress, memory, communication, healthy eating, and the management of sleep [20]. Home practice included creating 'action plans' that focussed on activities to enhance health and well-being. HSMP was delivered to groups of 7 to 13 participants by clinically trained facilitators with at least three years of experience in leading group-based clinical or psychoeducational interventions.

**Measures of well-being.** The 42-item *Psychological Well-being Scale* (PWBS; [14]) was used to measure psychological well-being as conceptualised by Ryff [12]. The PWBS is grounded in a theoretical model of psychological well-being that comprises six dimensions, namely self-acceptance, positive relations with others, autonomy (independence), environmental mastery (ability to manage life's demands), purpose in life, and personal growth (sense of developing and growing; [12]). Each dimension is measured by a 7-item subscale using a 7-point Likert scale ranging from 1 (strongly agree) to 7 (strongly disagree). After reverse scoring 21 items, subscale scores were derived by averaging their respective item scores; a total score was derived by averaging all items. Higher scores reflect higher levels of psychological well-being. The subscales of the PWBS have displayed low to moderate levels of internal consistency (Cronbach's alpha ranging from 0.33 to 0.56; [14]).

The psychological domain of the *World Health Organization WHOQOL-BREF Quality of Life Assessment* [13] was used to measure psychological quality of life. The WHOQOL Group conceptualises quality of life as a subjective evaluation of one's position in life in relation to the goals, expectations, and concerns that emerge from one's cultural, social, and environmental context. The psychological subscale of the WHOQOL-BREF captures levels of positive feelings (e.g., sense of meaningfulness) and body image, self-esteem, the ability to concentrate, and the lack of negative feelings (e.g., anxiety). The 6-item psychological subscale uses a 5-point Likert scale anchored at 0 (not at all) and 5 (completely). After reverse scoring one item, psychological subscale scores were derived by summing the six item scores. Higher scores are indicative of higher levels of psychological quality of life. The psychological subscale of the WHOQOL-BREF has displayed good levels of internal consistency (Cronbach's alpha = 0.81; [13]).

Three composite scores were used to measure the meditation-related well-being dimensions of awareness, connection, and insight as introduced by Dahl et al. [7]. In this framework, awareness describes a heightened and malleable attentiveness to perceptions (e.g., thoughts, feelings, and sensations) and a capacity to notice when levels of awareness decrease and the likelihood to be distracted increases. Connection describes a sense of care toward others that supports positive interactions and relationships. Connection encompasses feelings of gratitude, appreciation, and kinship, and a heightened capacity to understand and empathise with others' perspectives. Insight describes the capacity to experientially understand the ways in which thoughts, feelings, assumptions, and worldviews shape and participate in one's perception of self, other, and world. Awareness, connection, and insight correspond to the

attentional, constructive, and deconstructive psychological capacities previously introduced by Dahl et al. [16]. Details of the theory-guided development and psychometric properties of the composites used in the present study have been published [15]. The three composite scores include scales or subscales from four self-report measures (see Table 2): The *Multidimensional Assessment of Interoceptive Awareness* (MAIA; [21]) questionnaire and the *39-item Five Facet Mindfulness Questionnaire* (FFMQ-39; [22]) subscales of observing (noticing experiences) and acting with awareness (attending to activities non-mechanically) were used as measures of awareness. The *Compassionate Love Scale* (stranger-humanity version; [23, 24]) was used as a measure of connection. The *Drexel Defusion Scale* [25] and the FFMQ subscales of non-judging (non-evaluative stance towards experiences) and non-reactivity (allowing experiences) were used as measures of insight. Detailed descriptions of the self-report measures included in the composite scores can be found in S1 Table in S1 File.

To derive the three scores of meditation-related dimensions of well-being, we subtracted each scale score at each time point from the baseline pooled mean. We then divided this difference by the baseline pooled standard deviation. Next, each score was computed by averaging the z-scores of the scales that were assigned to the respective composite, yielding three composite scores with a baseline mean of 0 and a standard deviation smaller than one. Finally, to ease longitudinal data interpretation, we re-standardised each composite score so that longitudinal changes in each composite score reflect changes in standard deviation units.

## Statistical analyses

**Sample size.** Sample size calculations in SCD-Well were based on the expected effect size (0.5, based on a meta-analysis of the efficacy of meditation-based interventions for reducing anxiety symptoms; [26]) with 80% power and two-sided type I error of 5% for the mean change in trait-STAI scores from V1 to V2 between CMBAS and HSMP, resulting in a minimum total number of 128 (64 per group), which has been exceeded (n = 147; detailed in [18].

**Comparative analyses.** To assess between-group differences in mean changes in outcomes, we built one mixed effects linear regression model for each outcome incorporating data from all time points with an interaction term between visit and group. In all analyses, positive (negative) estimated mean between-group differences reflect higher (lower) changes in outcome scores in CMBAS. In accordance with the pre-registered statistical analysis plan for secondary outcomes of the Medit-Ageing Project, in all mixed effects regression models, missing data of the well-being outcomes were not replaced and assumed to be missing-at-random. The data and analysis plan underlying this paper are made available on request following approval by the executive committee and a formal data sharing agreement (https://silversantestudy.eu/2020/09/25/data-sharing). No participant data were excluded based on very high or low scale scores. Primary analyses of PWBS total scores, psychological QoL, and composite scores (awareness, connection, insight) were adjusted for multiple comparison (Bonferroni correction for multiple testing). Exploratory analyses of PWBS subscales were not adjusted for multiple comparison.

To test the potential moderating effect on measures of well-being within both groups, we built linear regression models with change in well-being scores from V1 to V2 as the outcome and the moderator variables of interest as the predictors. These variables included session attendance (out of a maximum of nine sessions, i.e. 8 weekly meetings plus one retreat day), baseline neuroticism measured by the neuroticism subscale of the 44-item Big Five Inventory [27], and baseline scores of the well-being outcomes. Analyses were conducted in R version 4.0.2 and Stata/MP version 16.0.

**Table 1. Baseline demographic characteristics.**

|  | CMBAS (n = 73) | HSMP (n = 74) |
|---|---|---|
| Age, in years | 72.1 (7.5) | 73.2 (6.2) |
| Female, n (%) | 47 (64.4%) | 48 (64.9%) |
| Education, in years | 13.9 (3.8) | 13.4 (3.4) |
| Ethnicity (white), n (%) | 69 (95%) | 72 (99%) |

*Note.* All variables are mean (standard deviation) unless otherwise specified. CMBAS = Caring Mindfulness-based Approach for Seniors; HSMP = Health Self-Management Programme.

## Results

Demographic characteristics are reported in Table 1. Descriptive statistics of well-being outcomes (based on all available data) are displayed in Table 2 and Fig 1. Results from mixed effects regression models assessing differential change in well-being outcomes (based on all participants who provided data at V1, V2, and V3) are shown in Table 3. There were no significant differences between the interventions for the mean number of sessions attended (CMBAS = 6.7; HSMP = 6.8), the proportion of participants who attended at least four sessions

**Table 2. Descriptive statistics for well-being outcomes by group and visit based on all available data.**

|  | CMBAS | | | | | | HSMP | | | | | |
|---|---|---|---|---|---|---|---|---|---|---|---|---|
|  | V1 | | V2 | | V3 | | V1 | | V2 | | V3 | |
| Outcome | n | Mean (SD) | n | Mean (SD) | n | Mean (SD) | n | Mean (SD) | n | Mean (SD) | n | Mean (SD) |
| PWBS |  |  |  |  |  |  |  |  |  |  |  |  |
| Total | 72 | 4.5 (1.2) | 59 | 4.4 (1.3) | 58 | 4.5 (1.2) | 70 | 4.5 (1.2) | 56 | 4.6 (1.3) | 63 | 4.6 (1.2) |
| Autonomy | 71 | 4.7 (1) | 59 | 4.9 (1) | 59 | 4.8 (0.9) | 70 | 4.9 (0.9) | 56 | 5.1 (0.9) | 63 | 5.1 (0.9) |
| Environmental mastery | 72 | 4.6 (1.5) | 59 | 4.5 (1.7) | 59 | 4.6 (1.6) | 70 | 4.5 (1.5) | 57 | 4.5 (1.6) | 63 | 4.6 (1.5) |
| Personal growth | 72 | 4.4 (1.3) | 59 | 4.3 (1.3) | 58 | 4.3 (1.3) | 70 | 4.3 (1.3) | 56 | 4.4 (1.4) | 63 | 4.3 (1.3) |
| Positive relations | 71 | 4.7 (1.5) | 59 | 4.5 (1.6) | 59 | 4.5 (1.6) | 70 | 4.8 (1.5) | 56 | 4.8 (1.6) | 63 | 4.8 (1.5) |
| Purpose in life | 72 | 4.3 (1.4) | 59 | 4.2 (1.5) | 58 | 4.3 (1.4) | 70 | 4.2 (1.4) | 56 | 4.2 (1.4) | 63 | 4.3 (1.4) |
| Self-acceptance | 72 | 4.4 (1.4) | 59 | 4.3 (1.5) | 59 | 4.4 (1.4) | 70 | 4.5 (1.5) | 56 | 4.5 (1.5) | 63 | 4.5 (1.5) |
| Psychological QoL | 71 | 21.6 (3.8) | 59 | 22.3 (3.8) | 59 | 22.2 (4.7) | 69 | 22.3 (3.1) | 58 | 22.7 (3.5) | 62 | 22.9 (3.4) |
| Awareness |  |  |  |  |  |  |  |  |  |  |  |  |
| MAIA noticing | 72 | 3.0 (1.2) | 59 | 3.1 (1.2) | 59 | 3.2 (1.2) | 69 | 2.8 (1.2) | 58 | 3.1 (1.2) | 61 | 2.9 (1.3) |
| MAIA attention regulation | 71 | 2.6 (1.1) | 59 | 2.8 (1.0) | 59 | 2.7 (0.9) | 67 | 2.8 (0.9) | 56 | 2.8 (0.9) | 62 | 2.9 (0.8) |
| MAIA emotional awareness | 72 | 3.3 (1.1) | 59 | 3.3 (1.1) | 59 | 3.3 (1.1) | 67 | 3.5 (1.0) | 58 | 3.4 (1.1) | 62 | 3.4 (1.1) |
| MAIA self-regulation | 71 | 2.4 (1.1) | 59 | 3.0 (1.1) | 59 | 3.0 (1.0) | 66 | 2.7 (1.0) | 57 | 2.9 (1.0) | 62 | 2.8 (1.0) |
| MAIA body listening | 71 | 1.8 (1.2) | 59 | 2.4 (1.0) | 58 | 2.3 (1.0) | 69 | 1.9 (1.2) | 58 | 2.1 (1.2) | 62 | 2.2 (1.1) |
| FFMQ observing | 72 | 9.6 (2.6) | 59 | 9.3 (2.2) | 60 | 9.2 (2.6) | 70 | 9.5 (2.7) | 58 | 9.6 (2.7) | 62 | 9.6 (2.8) |
| FFMQ act with awareness | 72 | 10.3 (3.0) | 59 | 10.2 (3.1) | 59 | 10.0 (3.0) | 70 | 10.6 (2.6) | 58 | 10.5 (2.9) | 62 | 11.0 (2.9) |
| Connection |  |  |  |  |  |  |  |  |  |  |  |  |
| Compassionate Love Scale | 71 | 89.5 (22.1) | 58 | 89.2 (21.5) | 59 | 86.9 (23.1) | 70 | 95.2 (18.1) | 58 | 92.2 (24.7) | 62 | 90.0 (22.5) |
| Insight |  |  |  |  |  |  |  |  |  |  |  |  |
| Drexel Defusion Scale | 71 | 30.1 (8.3) | 59 | 31.7 (9.7) | 60 | 32.7 (8.3) | 69 | 33.4 (8.4) | 58 | 34.6 (6.7) | 62 | 34.2 (7.0) |
| FFMQ non-judging | 72 | 11.8 (2.8) | 59 | 12.1 (2.6) | 60 | 11.7 (3.0) | 70 | 11.8 (2.5) | 58 | 12.1 (2.6) | 62 | 11.7 (3.0) |
| FFMQ non-reactivity | 72 | 9.6 (2.9) | 59 | 9.4 (3.1) | 59 | 9.6 (2.5) | 70 | 9.3 (3) | 58 | 9.1 (2.9) | 62 | 9.0 (2.8) |

*Note.* PWBS = Psychological Well-being Scale; QoL = Quality of Life; SD = standard deviation; CMBAS = Caring Mindfulness-based Approach for Seniors; HSMP = Health Self-Management Programme; PWBS = Psychological Well-being Scale; QoL = quality of life; MAIA = Multidimensional Assessment of Interoceptive awareness; FFMQ = Five Facet Mindfulness Questionnaire.

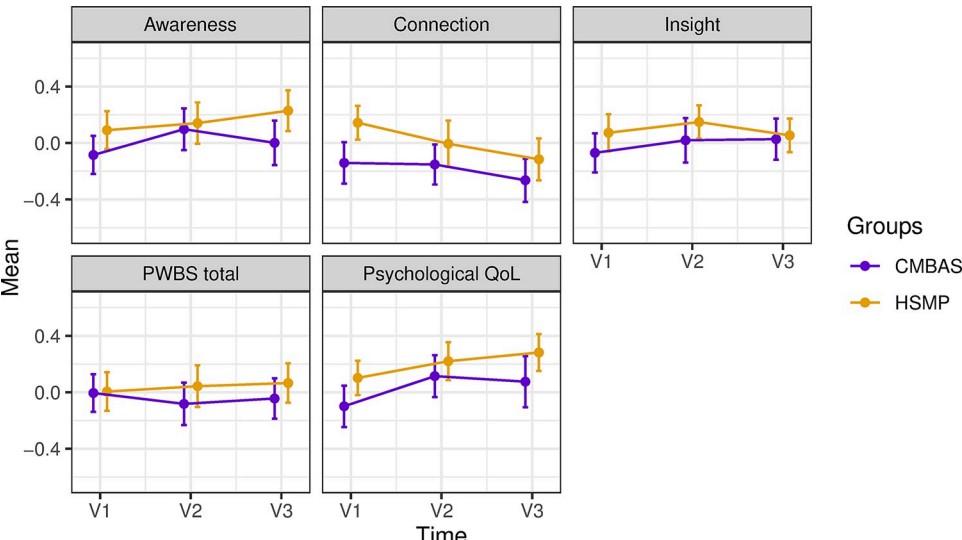

**Fig 1. Longitudinal changes in meditation-based well-being composite scores (awareness, connection, insight), Psychological Well-being Scale (PWBS) total scores, and WHOQOL-BREF Psychological Quality of Life (QoL) by group (CMBAS = Caring Mindfulness-based Approach for Seniors, HSMP = Health Self-Management Programme).** The figure displays observed standardised means and SEs (error bars = 1 SE) based on all available data.

(CMBAS = 81%; HSMP = 85%), or the proportion of participants who reported continued engagement with intervention activities between V2 and V3 (CMBAS = 59%; HSMP = 54%). There were no significant differences between the proportions of participants who completed home practice on at least four occasions (CMBAS = 55 [75%]; HSMP = 51 [69%]).

## PWBS

CMBAS and HSMP did not increase PWBS total scores, and no differences were observed between CMBAS and HSMP on changes in PWBS total scores (Table 3).

**Table 3. Results from mixed effects models assessing differential change in well-being outcomes.**

| Outcome | Time | Standardised estimated change | | Difference in change CMBAS vs. HSMP | |
|---|---|---|---|---|---|
| | | CMBAS | HSMP | Mean (95% CI) | p |
| PWBS total | V1 to V2 | 0.02 (-0.11, 0.14) | 0.05 (-0.07, 0.17) | 0.03 (-0.11, 0.18) | 0.638 |
| | V1 to V3 | 0.01 (-0.12, 0.13) | 0.09 (-0.03, 0.21) | 0.08 (-0.06, 0.23) | 0.253 |
| Psychological QoL | V1 to V2 | 0.18 (-0.06, 0.43) | 0.04 (-0.21, 0.29) | 0.14 (-0.15, 0.44) | 0.337 |
| | V1 to V3 | 0.09 (-0.35, 0.17) | 0.10 (-0.15, 0.34) | -0.002 (-0.29, 0.29) | 0.990 |
| Awareness | V1 to V2 | 0.17 (-0.07, 0.40) | 0.10 (-0.15, 0.35) | 0.08 (-0.22, 0.36) | 0.628 |
| | V1 to V3 | 0.05 (-0.19, 0.29) | 0.14 (-0.10, 0.38) | -0.08 (-0.37, 0.20) | 0.556 |
| Connection | V1 to V2 | 0.20 (-0.02, 0.42) | -0.18 (-0.40, 0.04) | **0.38** (0.12, 0.64) | 0.004 |
| | V1 to V3 | -0.01 (-0.22, 0.21) | -0.31 (-0.53, -0.10) | 0.30 (0.05, 0.56) | 0.020 |
| Insight | V1 to V2 | 0.12 (-0.10, 0.35) | 0.02 (-0.21, 0.25) | 0.10 (-0.16, 0.37) | 0.454 |
| | V1 to V3 | 0.10 (-0.12, 0.33) | -0.04 (-0.26, 0.18) | 0.14 (-0.12, 0.41) | 0.284 |

*Note.* Only participants who provided data at all three time points were included in the analyses. All analyses were adjusted for baseline scores of the outcome. Estimates in bold were associated $p < 0.005$ (significance threshold adjusted using the Bonferroni correction for multiple testing). PWBS = Psychological Well-being Scale; QoL = Quality of Life; SCD = subjective cognitive decline; CI = confidence interval; CMBAS = Caring Mindfulness-based Approach for Seniors; HSMP = Health Self-Management Programme.

Exploratory analyses indicated that across PWBS dimensions, only PWBS autonomy increased within both groups from V1 to V2 (CMBAS: Cohen's d: 0.22 [95% CI: 0.02, 0.42], p = 0.023; HSMP: Cohen's d: 0.24 [95% CI: 0.03, 0.44], p = 0.018) and from V1 to V3 in HSMP only (Cohen's d: 0.22 [95% CI: 0.02, 0.41], p = 0.026; S2 Table in S1 File). Neither CMBAS nor HSMP increased other PWBS dimensions from V1 to V2 or from V1 to V3.

## Psychological QoL

No differences were observed between CMBAS and HSMP on changes in psychological QoL from V1 to V2 (Cohen's d: 0.15 [95% CI: -0.08, 0.37], p = 0.206) and from V1 to V3 (Cohen's d: 0.15 [95% CI: -0.08, 0.37], p = 0.206). No within-group changes were found.

## Meditation-based well-being dimensions

CMBAS was superior to HSMP on changes in connection from V1 to V2 (Cohen's d: 0.38 [95% CI: 0.12, 0.64], p = 0.004). From V1 to V2, connection did not change within CMBAS (Cohen's d: 0.20 [95% CI: -0.02, 0.42], p = 0.082) or within HSMP (Cohen's d: -0.18 [95% CI: -0.40, 0.04], p = 0.132). From V1 to V3, a significant decrease in connection was observed within HSMP (Cohen's d: -0.31 [95% CI: -0.53, -0.10], p = 0.002). No differences were observed between CMBAS and HSMP on changes in awareness and insight (all p-values > 0.284), and no within-group changes were observed for these outcomes.

## Moderator analyses

Exploratory moderator analyses were conducted within both groups to assess the association between baseline characteristics and intervention response (i.e., from V1 to V2). For a selected number of outcomes (CMBAS: awareness, connection, insight, psychological QoL; HSMP: insight), higher baseline scores were associated with weaker improvements. Neuroticism did not moderate the effects of CMBAS or HSMP. Session attendance showed a moderating effect on connection, with higher session attendance predicting a greater decrease in connection in CMBAS and a greater increase in connection in HSMP. All moderator analyses can be found in S3 Table in S1 File.

# Discussion

Utilising three theory-based conceptions of well-being [7, 12, 13] in this large, multinational clinical trial of older adults with SCD, an 8-week CMBAS and a structurally matched HSMP were associated with only limited effects on psychological well-being. CMBAS was superior to HSMP on changes in connection at post-intervention. Within both groups, PWBS total scores, psychological QoL, and composite scores did not increase significantly from baseline to post-intervention or follow-up. Exploratory analyses suggested that levels of autonomy increased within both groups during the intervention. In Ryff's framework of well-being [14], increasing levels of autonomy reflect an increased capacity to be independent, self-determined, and able to view oneself and regulate one's behaviour based on personal standards rather than social and cultural pressures.

Overall, however, our findings contrast with our hypotheses. Based on previous research and theory [7, 17, 28], we expected CMBAS to positively impact various dimensions of psychological well-being and human flourishing. The primary outcome of the SCD-Well trial was mean change in levels of trait anxiety from pre- to post-intervention [18]. Within both CMBAS and HSMP, trait anxiety was reduced in statistically significant and clinically meaningful ways [9]. The magnitude of these effects on the primary outcome did not fully translate

to the well-being measures presented here. Despite decreases in trait anxiety, CMBAS' limited effects on psychological well-being raise concerns about the utility and specificity of an 8-week MBI in older adults with SCD.

Several potential explanations for these unexpected findings can be considered. For instance, one explanation relates to the limitations of the well-being measures we employed. The PWBS [14] and WHOQOL-BREF Quality of Life Assessment [13] were not informed by contemplative perspectives or developed to measure the effects of meditation training. These well-being measures might be limited in their ability to capture those dimensions of well-being that meditation theories would predict long-term practice to cultivate [7, 16]. In fact, a recent cross-sectional study suggested that expert meditators ($\geq$10,000 hours of practice including one 3-year meditation retreat) displayed lower PWBS total scores than meditation-naïve individuals [15]. Nonetheless, from a clinical perspective, we still expected an improvement in the general type of well-being that is captured by these measures. Importantly, the present study did include composite measures that were theoretically derived from meditation-based dimensions of well-being (i.e., awareness, connection, insight; [7]). Although the impact of CMBAS on awareness and insight was arguably trending towards a meaningful effect size post-intervention, this impact was not detectable anymore at the 6-month follow-up. Another explanation for the limited effects on psychological well-being could be related to the length of the meditation training period. Although 8-week MBIs in younger healthy populations have exerted a positive impact on measures of global well-being as well as dimensions of awareness, connection, and insight (e.g., [29]), in older adults with SCD, eight weeks of practice might be too brief for measurable and clinically meaningful changes in facets of psychological well-being to manifest. Notably, in MBIs in younger healthy populations, effect sizes of change in measures of psychological distress tend to be larger than those of changes in well-being dimensions [29, 30]. This pattern also emerges in the context of the SCD-Well trial and is in line with the fact that standard MBIs, derived from the generic mindfulness-based stress reduction programme, are mainly targeted at helping participants develop more adaptive responses to psychological distress. One potential explanation for this pattern is that greater intervention duration is required for psychological well-being to improve than for psychological distress (e.g., anxiety) to decrease. In that regard, a potential lack of statistical power could have also contributed to the limited effects on well-being as the SCD-Well trial was designed to primarily detect effects on levels of trait anxiety [18]. Further, the limited intervention effects could also be related to factors that have been associated with SCD but were not sufficiently captured in the context of the present study (e.g., sleep disturbances measured by polysomnography). Longitudinal studies with longer training periods and additional measures of physical and mental health are needed to further elucidate these questions and other potential dose-response relationships between meditation practice and diverse aspects of psychological well-being in older adults. The ongoing Age-Well trial [31], which includes the longest meditation training conducted to date and utilises similar measures of well-being to the present study, could contribute to begin answering these questions.

Trajectories of change in outcomes might vary substantially depending on participants' baseline characteristics; yet only few moderators of meditation training have been consistently found or considered [32]. Previous work has suggested that individuals who display better/poorer functioning at baseline might show a smaller/larger response to meditation-based interventions (see [33]). For individuals who are relatively psychologically well at baseline, longer training periods might be required to achieve noticeable levels of improvement. Here, we evaluated the moderating effects of baseline levels of neuroticism (i.e., proneness to experience distress) and well-being. In line with prior predictions, higher levels of awareness, connection, insight, and psychological QoL at baseline were associated with smaller improvements post-

CMBAS. The opposite pattern in which higher baseline scores predicted stronger intervention response was not found for any outcome. Baseline scores of neuroticism did not predict participants' response to CMBAS. Further, session attendance showed no moderating effects on well-being outcomes except on connection, with higher session attendance predicting, unexpectedly, a greater decrease in connection. Given the exploratory nature of these moderation analyses and the lack of prior studies investigating the effects of MBIs on well-being in patients with SCD, we hesitate to offer explanations for this counterintuitive moderation finding.

The SCD-Well trial has important strengths. Aiming to address several previously-identified limitations of meditation research [7, 33, 34], the trial included a theory-based active comparison condition; the mindfulness-based intervention was based on a tailored, manualised training paradigm that was informed by the strengths and limitations of previous work; we utilised theoretical models of meditation practice that were informed by psychological, neuroscientific, and contemplative perspectives [16]; we compared established scientific models of psychological well-being to a recent meditation-based framework for human flourishing [7]; and we addressed the need for studies of meditation-based interventions in older adults (see [32]).

The trial also has important limitations. The generalisability of our findings to other populations of older adults is reduced because our sample consisted of well-educated and largely white participants. Further, we did not include a passive control group to assess fluctuations in wellbeing independent of the interventions. Importantly, no self-report measures that specifically reflect the dimensions of Dahl et al.'s training-based framework for well-being [7] have been developed. Therefore, we utilised previously developed composite scores of meditation-related capacities that were based on self-report measures of trait-like individual differences [15]. These trait-level scales may be suboptimal for capturing the process-level aspects of meditation-related dimensions of psychological well-being.

## Supporting information

**S1 File. Contains supporting tables.**
(PDF)

## Acknowledgments

The Medit-Ageing Research Group includes: Claire André, Nicholas Ashton, Florence Allais, Julien Asselineau, Eider Arenaza-Urquijo, Romain Bachelet, Martine Batchelor, Axel Beaugonin, Viviane Belleoud, Clara Benson, Beatriz Bosch, Maelle Botton, Maria Pilar Casanova, Pierre Champetier, Anne Chocat, Nina Coll, Sophie Dautricourt, Pascal Delamillieure, Vincent De La Sayette, Marion Delarue, Harriet Demnitz-King, Titi Dolma, Stéphanie Egret, Francesca Felisatti, Eglantine Ferrand-Devouges, Eric Frison, Francis Gheysen, Karine Goldet, Julie Gonneaud, Abdul Hye, Agathe Joret Philippe, Elizabeth Kuhn, Brigitte Landeau, Gwendoline Ledu, Valérie Lefranc, Maria Leon, Dix Meiberth, Florence Mezenge, Ester Milz, Inès Moulinet, Hendrik Mueller, Theresa Mueller, Valentin Ourry, Cassandre Palix, Léo Paly, Géraldine Poisnel, Anne Quillard, Alfredo Ramirez, Géraldine Rauchs, Florence Requier, Leslie Reyrolle, Ana Salinero, Eric Salmon, Lena Sannemann, Yamna Satgunasingam, Christine Schwimmer, Hilde Steinhauser, Edelweiss Touron, Denis Vivien, Patrik Vuilleumier, Cédrick Wallet, Tim Whitfield, and Janet Wingrove. Many people helped in implementing these projects. The authors would like to thank all the contributors listed in the Medit-Ageing Research Group as well as Rhonda Smith, Charlotte Reid, the sponsor (Pôle de Recherche Clinique at Inserm), Inserm Transfert (Delphine Smagghe), and the participants in the Medit-Ageing

project. Please address any correspondence relating to the Medit-Ageing Research Group to the project lead Gaël Chételat (chetelat@cyceron.fr).

## Author Contributions

**Conceptualization:** Marco Schlosser, Thorsten Barnhofer, Fabienne Collette, Julie Gonneaud, Gaël Chételat, Frank Jessen, Olga M. Klimecki, Antoine Lutz, Natalie L. Marchant.

**Data curation:** Marco Schlosser.

**Formal analysis:** Marco Schlosser.

**Funding acquisition:** Gaël Chételat.

**Investigation:** Marco Schlosser, Harriet Demnitz-King.

**Methodology:** Marco Schlosser, Thorsten Barnhofer, Fabienne Collette, Gaël Chételat, Olga M. Klimecki, Antoine Lutz, Natalie L. Marchant.

**Supervision:** Matthias Kliegel, Olga M. Klimecki, Antoine Lutz, Natalie L. Marchant.

**Validation:** Antoine Lutz.

**Visualization:** Marco Schlosser.

**Writing – original draft:** Marco Schlosser.

**Writing – review & editing:** Marco Schlosser, Harriet Demnitz-King, Thorsten Barnhofer, Fabienne Collette, Julie Gonneaud, Gaël Chételat, Frank Jessen, Matthias Kliegel, Olga M. Klimecki, Antoine Lutz, Natalie L. Marchant.

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
