## [Decision Letter · Decision Letter 0]

24 Aug 2023

PONE-D-23-18446Effects of a mindfulness-based intervention and a health self-management programme on psychological well-being in older adults with subjective cognitive decline: Secondary analyses from the SCD-Well randomised clinical trialPLOS ONE

Dear Dr. Lutz,

Thank you for submitting your manuscript to PLOS ONE. After careful consideration, we feel that it has merit but does not fully meet PLOS ONE’s publication criteria as it currently stands. Therefore, we invite you to submit a revised version of the manuscript that addresses the points raised during the review process.

ACADEMIC EDITOR:Dear Authors,

Please, respond to reviewers recommendations in order to submit the new version of the paper (incl. the responses to reviewers).

Best regards

Editor

We look forward to receiving your revised manuscript.

Kind regards,

Marcelo Marcos Piva Demarzo, MD, PhD

Academic Editor

PLOS ONE

“GC has received research support from the EU’s Horizon 2020 research and innovation programme (grant agreement number 667696), Inserm, Fondation d’entreprise MMA des Entrepreneurs du Futur, Fondation Alzheimer, Programme Hospitalier de Recherche Clinique, Région Normandie, Association France Alzheimer et maladies apparentées and Fondation Vaincre Alzheimer (all to Inserm), and personal fees from Fondation d’entreprise MMA des Entrepreneurs du Futur. All other authors have declared that no competing interests exist.”

3. One of the noted authors is a group or consortium [The Medit-Ageing Research Group]. In addition to naming the author group, please list the individual authors and affiliations within this group in the acknowledgments section of your manuscript. Please also indicate clearly a lead author for this group along with a contact email address.

Additional Editor Comments:

Dear Authors,

Please, respond to reviewers recommendations in order to submit the new version of the paper (incl. the responses to reviewers).

Best regards

Editor

Reviewers' comments:

Reviewer's Responses to Questions

**Comments to the Author**

1. Is the manuscript technically sound, and do the data support the conclusions?

Reviewer #1: Yes

Reviewer #2: Yes

2. Has the statistical analysis been performed appropriately and rigorously? 

Reviewer #1: I Don't Know

Reviewer #2: I Don't Know

3. Have the authors made all data underlying the findings in their manuscript fully available?

Reviewer #1: Yes

Reviewer #2: No

4. Is the manuscript presented in an intelligible fashion and written in standard English?

Reviewer #1: Yes

Reviewer #2: Yes

5. Review Comments to the Author

Reviewer #1: Important note: This review pertains only to ‘statistical aspects’ of the study and so ‘clinical aspects’ [like medical importance, relevance of the study, ‘clinical significance and implication(s)’ of the whole study, etc.] are to be evaluated [should be assessed] separately/independently. Further please note that any ‘statistical review’ is generally done under the assumption that (such) study specific methodological [as well as execution] issues are perfectly taken care of by the investigator(s). This review is not an exception to that and so does not cover clinical aspects {however, seldom comments are made only if those issues are intimately / scientifically related & intermingle with ‘statistical aspects’ of the study}. Agreed that ‘statistical methods’ are used as just tools here, however, they are vital part of methodology [and so should be given due importance]. I look at the manuscript in/with statistical view point, other reviewer(s) look(s) at it with different angle so that in totality the review is very comprehensive. However, there should be efforts from authors side to improve (may be by taking clues from reviewer’s comments). Therefore, please do not limit the revision only (with respect) to comments made here.

COMMENTS: Although this manuscript is well drafted and the study seems to have enough potential (is on very important/useful topic), I have few doubts/observations/concerns (different opinion) which are little serious and are given below:

In ‘Methods’ section it is stated that ‘Detailed information about the recruitment procedures, eligibility criteria, design of the interventions, and assessments can be found in the trial protocol’ and quoted reference (18). But reference 18 seems to be a report of the trial and not the ‘PROTOCOL’. It is stated there that ‘The treatment is based on a published manual [24], with every session of the program covering different subjects, including self-management’ which indicates that some sort of / some amount of ‘self-management’ component was there. Then is it not necessary to clarify regarding the overlap? It is well-known (NET/WWW search) that “Secondary analysis refers to the use of existing research data to find answer to a question that was different from the original work” then is this study can be called as ‘Secondary analyses’ [as mentioned in title]? I am sure that the authors are aware of limitations/disadvantages of Secondary Data Analysis, however, I request authors to kindly read the following pasted from one famous standard textbook on ‘Medical Research Methodology’:

Since the researcher did not collect the data, he or she has no control over what is contained in the data set. Often times this can limit the analysis or alter the original questions the researcher sought out to answer.

You may consider to change the ‘title’ (only if convinienced). The present study (manuscript) seems to be a report of component/part [hitherto unpublished in this form] of the same (the SCD-Well randomised controlled trial of the European) trial. Further, please note that though application of ‘Mixed effects models’ here is perfectly alright, they are [any regression techniques for that matter] are not basically/originally developed for any sort of [between or within group(s)] comparison(s). In ‘Methods’ section you stated that “Mixed effects models were used to assess between- and within-group differences in change”. Head-to-head comparison is expected, as this is [through ‘Mixed effects models’] an indirect/secondary/by-product testing, in my opinion. Application of ‘Mixed effects models’ could definitely be (useful) addition (I am sure). Even if ultimate results are same, one should follow a correct way (I think).

Note that, though the measures/tools used are appropriate {example: 42-item Psychological Well-being Scale (PWBS), Quality of Life Assessment (by WHOQOL-BREF), four self-report measures (displayed in Table 2) - The Multidimensional Assessment of Interoceptive Awareness (MAIA) questionnaire and the 39-item Five Facet Mindfulness Questionnaire (FFMQ-39), etc.}, most of them are likely to yield data that are in [at the most] ‘ordinal’ level of measurement [and not in ratio level of measurement for sure {as the score two times higher does not indicate presence of that parameter/phenomenon as double (for example, a Visual Analogue Scales VAS score or say ‘depression’ score)}]. Then application of suitable non-parametric (or distribution free) test(s) is/are indicated/advisable [even if distribution may be ‘Gaussian’ (also called ‘normal’)]. Agreed that there is/are no non-parametric test(s)/technique(s) available to be used as alternative in all situation(s), but should be used whenever/wherever they are available. Therefore, in short use suitable non-parametric test(s)/technique(s) while dealing with data that are in ‘ordinal’ level of measurement even if [despite that] the distribution may be ‘Gaussian’. Testing ‘normality’ in sample [by using any normality test(s)} is not required/desired while dealing with data that are in ‘ordinal’ level of measurement [as most of the normality tests are not valid for ‘ordinal’ data].

It may also (as often said) please be noted, [as you stated: Each dimension is measured by a 7-item subscale using a 7-point Likert scale ranging from 1 (strongly agree) to 7 (strongly disagree). After reverse scoring 21 items, subscale scores were derived by averaging their respective item scores; a total score was derived by averaging all items] that whenever response options ranged from 1 (=strongly agree) to 7 (=strongly disagree) {or from 1=very bad to 3=neither good nor bad to 5=very good), while using a ‘Likert’ scale responses, recoding [like strongly disagree=-2, disagree=-1, neutral=0, agree=1, strongly agree=2] may yield correct and meaningful ‘arithmetic mean’ which is useful not only for comparison but has absolute meaning, in my opinion. Application of any statistical test(s) assume that meaning of entity used (mean, SD, etc) has a particular meaning. Though ‘α’ [alpha] or most other measures of reliability/correlation will remain same, however, use of non-parametric methods should/may be preferred while dealing with data yielded by any questionnaire/score.

At the end of ‘Introduction’ section it is said that “We hypothesised that both interventions would improve well-being but that CMBAS would be superior to HSMP”, then what about the principle of ‘equipoise’ [equipoise means that there is genuine uncertainty in the expert medical community over whether a treatment will be beneficial. An ethical dilemma arises in a clinical trial when the investigator(s) begin to believe that the treatment or intervention administered in one arm of the trial is significantly outperforming the other arms.]? Again, as often said, some sort of ‘bias’ is (are) likely be introduced/present when the principle of ‘equipoise’ is not observed/followed.

In ‘Statistical analyses -Sample size’ section is it adequate to say “Sample size calculations in SCD-Well were based on the expected effect size ….’? Do not it necessary to mention ‘what that expected effect size’ you are referring to?

Except these minor points, the article is acceptable [drafting is excellent]. Nevertheless, mind you that as pointed out in ‘important note’ above “This review pertains only to ‘statistical aspects’ of the study and so ‘clinical aspects’ should be assessed separately/independently. ‘Minor Revision’ is recommended.

Reviewer #2: This is a well-written manuscript reporting the findings of an RCT comparing the effects of a mindfulness-based intervention vs. a health education program on wellbeing-related outcomes in participants with subjective cognitive decline. I will defer to the statistical reviewer regarding the appropriateness and rigor of the statistical analysis. While the data is not being made fully available without restriction, it appears that the authors have addressed the issue satisfactorily. The following are items I believe still need to be addressed.

1. It appears that some items recommended by CONSORT were not reported. I suggest consulting the CONSORT NPT extension and addressing any items that are currently lacking. I believe PLOS ONE also requires a completed CONSORT checklist and flow diagram as part of the submission for manuscripts reporting results of clinical trials.

2. The citation style is inconsistent and should be corrected.

3. In describing the HSMP intervention a published manual is mentioned. A citation to the manual should be provided.

4. In the comparative analyses section a pre-registered statistical analysis plan is mentioned. If this plan is available, information on where to access it should be provided.

5. Session attendance is included as a potential moderator variable, yet home practice did not seem to be included. Given that the majority of the time spent doing the intervention appears to be during home practice, I believe it warrants attention. On a related note, information on adherence (e.g., number of sessions attended, number/hours of home practice completed) would also be useful.

6. Looking at table 2, in general it appears there was a drop off in the number of participants from V1 to V2 for both groups, then V3 rebounded a bit for the HSMP group but not the CMBAS group. Was there some reason that might explain this difference?

7. When discussing generalizability of the findings, it is mentioned that the participants were largely white. The data supporting this should be provided in Table 1.

6. PLOS authors have the option to publish the peer review history of their article (what does this mean?). If published, this will include your full peer review and any attached files.

Reviewer #1: No

Reviewer #2: No

---

## [Author Response · Author response to Decision Letter 0]

30 Aug 2023

Please find the response letter attached.

---

## [Decision Letter · Decision Letter 1]

10 Oct 2023

PONE-D-23-18446R1Effects of a mindfulness-based intervention and a health self-management programme on psychological well-being in older adults with subjective cognitive decline: Secondary analyses from the SCD-Well randomised clinical trialPLOS ONE

Dear Dr. Lutz,

Thank you for submitting your manuscript to PLOS ONE. After careful consideration, we feel that it has merit but does not fully meet PLOS ONE’s publication criteria as it currently stands. Therefore, we invite you to submit a revised version of the manuscript that addresses the points raised during the review process.

We look forward to receiving your revised manuscript.

Kind regards,

Marcelo Marcos Piva Demarzo, MD, PhD

Academic Editor

PLOS ONE

Journal Requirements:

Additional Editor Comments:

Dear Authors,

Thank you for sending the revised manuscript, which partially met the quality criteria for final acceptance.

For final acceptance of the manuscript to occur, please pay attention to the following minor points (raised by our reviewers):

- In the introduction, the authors indicated “Although the condition could be an indication of prodromal Alzheimer’s disease (AD), which is the most common form of dementia (3), SCD has also been related to other factors (e.g., physical and mental illness, sleep disturbances, personality traits, effects of drugs).” It could be that the lack of effect is due to these other factors, especially when there is no consensus of what constitutes an SCD. Please, address this point in the discussion;

- Note that was suggested by one reviewer that (refer to previous item 4) “Head-to-head comparison” could be ideally used instead of ‘Mixed effects models’. Please, answer this comment in more details, and, if necessary, point this as a limitation in the discussion section.

Best regards

Editor

Reviewers' comments:

Reviewer's Responses to Questions

**Comments to the Author**

1. If the authors have adequately addressed your comments raised in a previous round of review and you feel that this manuscript is now acceptable for publication, you may indicate that here to bypass the “Comments to the Author” section, enter your conflict of interest statement in the “Confidential to Editor” section, and submit your "Accept" recommendation.

Reviewer #1: All comments have been addressed

Reviewer #3: (No Response)

2. Is the manuscript technically sound, and do the data support the conclusions?

Reviewer #1: (No Response)

Reviewer #3: (No Response)

3. Has the statistical analysis been performed appropriately and rigorously? 

Reviewer #1: (No Response)

Reviewer #3: (No Response)

4. Have the authors made all data underlying the findings in their manuscript fully available?

Reviewer #1: (No Response)

Reviewer #3: (No Response)

5. Is the manuscript presented in an intelligible fashion and written in standard English?

Reviewer #1: (No Response)

Reviewer #3: (No Response)

6. Review Comments to the Author

Reviewer #1: COMMENTS: All the comments are answered and so I recommend the acceptance because the manuscript has now achieved the acceptable level.

[However, please note that I suggested (refer to item 4) “Head-to-head comparison” {and said comparative inference drawn through ‘Mixed effects models’ done here is an indirect/secondary/by-product testing, in my opinion. Application of ‘Mixed effects models’ could definitely be (useful) addition (I am sure). Even if ultimate results are same, one should follow a correct way (I think)]. In response it is said that “We agree with the reviewer that the mixed effects models we have employed are an appropriate and effective method to answer our research questions” and there is NO mention of “Head-to-head comparison”.]

Reviewer #3: In the introduction, the authors indicated “Although the condition could

be an indication of prodromal Alzheimer’s disease (AD), which is the most common form of

dementia (3), SCD has also been related to other factors (e.g., physical and mental illness,

sleep disturbances, personality traits, effects of drugs).” It could be that the lack of effect is due to these other factors, especially when there is no consensus of what constitutes an SCD. It will be prudent to address this issue in the discussion.

7. PLOS authors have the option to publish the peer review history of their article (what does this mean?). If published, this will include your full peer review and any attached files.

Reviewer #1: **Yes: **Dr. Sanjeev Sarmukaddam

Reviewer #3: No

---

## [Author Response · Author response to Decision Letter 1]

1 Nov 2023

Please find the response attached.

---

## [Editor Report · Decision Letter 2]

16 Nov 2023

Effects of a mindfulness-based intervention and a health self-management programme on psychological well-being in older adults with subjective cognitive decline: Secondary analyses from the SCD-Well randomised clinical trial

PONE-D-23-18446R2

Dear Dr. Marchant,

We’re pleased to inform you that your manuscript has been judged scientifically suitable for publication and will be formally accepted for publication once it meets all outstanding technical requirements.

Kind regards,

Marcelo Marcos Piva Demarzo, MD, PhD

Academic Editor

PLOS ONE